# Accurate Building Extraction from Fused DSM and UAV Images Using a Chain Fully Convolutional Neural Network



**Wei Liu [1,2,\*]**, **MengYuan Yang [1]**, **Meng Xie [1]**, **Zihui Guo [1]**, **ErZhu Li [1]**, **Lianpeng Zhang [1]**, **Tao Pei [2]** and **Dong Wang [3]**

1. School of Geography, Geomatics and Planning, Jiangsu Normal University, Xu Zhou 221116, China; 2020170743@jsnu.edu.cn (M.Y.); 2020170774@jsnu.edu.cn (M.X.); guozihui@jsnu.edu.cn (Z.G.); liezrs2018@jsnu.edu.cn (E.L.); zhanglp@jsnu.edu.cn (L.Z.)
2. State Key Laboratory of Resources and Environmental Information System, Institute of Geographic Sciences and Natural Resources Research, CAS, Beijing 100101, China; peit@lreis.ac.cn
3. Zhejiang Design Institute of Water Conservancy & Hydro-electric Power, Hangzhou 310002, China; wangd86@zjwater.gov.cn
* Correspondence: liuw@jsnu.edu.cn; Tel.: +86-130-1393-9855

**Abstract:** Accurate extraction of buildings using high spatial resolution imagery is essential to a wide range of urban applications. However, it is difficult to extract semantic features from a variety of complex scenes (e.g., suburban, urban and urban village areas) because various complex man-made objects usually appear heterogeneous with large intra-class and low inter-class variations. The automatic extraction of buildings is thus extremely challenging. The fully convolutional neural networks (FCNs) developed in recent years have performed well in the extraction of urban man-made objects due to their ability to learn state-of-the-art features and to label pixels end-to-end. One of the most successful FCNs used in building extraction is U-net. However, the commonly used skip connection and feature fusion refinement modules in U-net often ignore the problem of feature selection, and the ability to extract smaller buildings and refine building boundaries needs to be improved. In this paper, we propose a trainable chain fully convolutional neural network (CFCN), which fuses high spatial resolution unmanned aerial vehicle (UAV) images and the digital surface model (DSM) for building extraction. Multilevel features are obtained from the fusion data, and an improved U-net is used for the coarse extraction of the building. To solve the problem of incomplete extraction of building boundaries, a U-net network is introduced by chain, which is used for the introduction of a coarse building boundary constraint, hole filling, and "speckle" removal. Typical areas such as suburban, urban, and urban villages were selected for building extraction experiments. The results show that the CFCN achieved recall of 98.67%, 98.62%, and 99.52% and intersection over union (IoU) of 96.23%, 96.43%, and 95.76% in suburban, urban, and urban village areas, respectively. Considering the IoU in conjunction with the CFCN and U-net resulted in improvements of 6.61%, 5.31%, and 6.45% in suburban, urban, and urban village areas, respectively. The proposed method can extract buildings with higher accuracy and with clearer and more complete boundaries.

**Keywords:** building extraction; digital surface model; unmanned aerial vehicle images; chain full convolution neural network; fusion

## 1. Introduction

Automatic extraction of building information while using unmanned aerial vehicles (UAVs), aerospace, and remote sensing satellite imagery is an important component in many fields, including

illegal land use monitoring, land-use change, image interpretation, and cartography [1,2]. Dividing pixels into semantic objects is one of the most challenging and important issues in urban aviation and satellite imagery. This is because high spatial resolution imagery usually have complex data features, and the features often appear in heterogeneous forms with large intra-class and low inter-classes variations, which are more prominent in buildings [3]. This heterogeneity of remote sensing images limits most traditional building extraction methods, which rely on a series of predefined features that are extracted by tunable parameters. Therefore, designing an automatic building extraction method that can achieve high precision and robustness is of utmost interest in optical remote sensing applications.

Accurate, reliable, and robust automatic extraction of buildings is still a huge challenge in remote sensing image processing, although tremendous progress has been made on various building extraction methods in recent decades [4]. The following factors present difficulties in building extraction: (1) buildings have different shapes, sizes, and spectrum reflectance in most scenes [5], (2) especially in suburban and urban village areas, buildings are often shaded by tall trees and their shadows, and (3) in high spatial resolution remote sensing imagery, due to the existence of spectrally similar pixels, buildings have the characteristics of high intra-class variance and low inter-class variance, which makes it difficult to extract the texture and spatial geometric features of buildings [3–6].

Texture feature, building index, contour tracking, and perceptual grouping are among the traditional methods for building extraction [7–11], The main limitations of these methods are their dependence on the features of input images and their inefficiency in processing huge amounts of input datasets [12]. Some machine learning models, such as support vector machine (SVM) and artificial neural networks (ANNs), overcome the limitations of traditional building extraction methods and successfully solve several building extraction problems [1,13,14]. It is worth noting that traditional machine learning methods (such as SVM and ANNs) mainly enhance semantic information through a large number of input features, thus reducing the ambiguity in buildings with spectral/geometric similarity and other man-made objects. Feature extraction is a time-consuming and laborious process, which requires detailed engineering design and expertise. This is because, in a particular problem, the efficiency of each feature is unknown and it needs further verification [15]. Moreover, these hand-crafted low-rise features have inferior generalization ability in distinguishing between various buildings. This means that these low-rise features are data- and regionally-oriented, and, while they work effectively for a specific case, they are less efficient in some other conditions.

Recent researches show that deep convolutional neural networks (DCNNs) could reach an impressive advanced performance for scene classification [16,17], object detection [18–20], and semantic segmentation [6,21,22] while using remote sensing imagery. DCNNs can accurately extract semantic features not only the low-levels and middle-levels, but also the high-level features from the input image [23]. DCNNs can be used to conduct semantic segmentation for remote sensing imagery, where each pixel is marked with its classification. However, the above methods usually generate maps with low-resolution features than the input raw images and show imprecise results in pixel-level labeling [24]. Long et al. proposed a full convolutional neural network (FCN) that achieves state-of-the-art performance for both pixel-based image classification and semantic segmentation while using an encoder-decoder approach [25]. FCNs have now become the general framework for some of the most advanced classification and image segmentation tasks due to their ability to obtain full-resolution feature maps [26]. U-net is one of the most successful FCNs used in this work [27]. U-net, which was proposed in 2016 for medical image segmentation, is built on the basis of adding a skip connection to the FCNs between the encoder and decoder. With this skipping connection, the decoder can receive a low-level feature from the encoder and then create the output without losing the boundary information. In the process of building extraction, it is extremely important to preserve the correct boundary information, which means that U-net can be used to detect fine details of building in the input images. Therefore, U-net is selected as the basic network for building extraction in this study.

FCNs and U-net are widely used to extract various natural and man-made objects, for instance, buildings, airplanes, canal, roads, and disaster-stricken areas [28–31]. Mnih proposed a large scale deep convolution neural network while using high spatial resolution aerial images for building and

road extraction [32]. Rasha et al. established an end-to-end convolutional neural network (CNN) model to simultaneously extract buildings and roads from high spatial resolution optical imagery [3]. Xu et al. proposed a novel FCN architecture for building extraction and utilized both hand crafted features and post processing to improve the building extraction results [33]. Huang et al. developed an encoder-decoder gated residual refinement network (GRRNet) that combines high spatial resolution remote sensing imagery and secondary data for building extraction [6]. Wu et al. modified U-Net to both predict the building roof and the outline where the outline played a regulatory role [34]. Raveerat et al. established a new U-net-based network with a symmetrical eight layer encoder -decoder architecture to detect the new construction of buildings in developing areas between two SAR images taken at different times [31].

At present, some issues that are presented by FCNs and U-net are yet to be solved. First, FCNs use high-level but coarse resolution image features of DCNNs for pixel level classification. However, FCNs often produces "speckle" building extraction results, as rich low- and middle-level semantic features, such as building corners and edges, are largely ignored [35]. U-net has an encoder-decoder architecture that transfers the low- and middle-level features to the decoder part by some skipping connection that can solve the "speckle" problem to some extent [36]. However, in the process of building extraction, there are inevitably some deficiencies, such as coarse boundary, hole, and "speckle" phenomenon. Second, the features transmitted by U-net may contain classification ambiguity or non-boundary related information [27] that will have an impact on the classification results and building boundaries. It is easy to confuse the classification between man-made objects (e.g., roads and fences) and buildings [31]. Third, U-net focuses on building extraction from high spatial resolution images and their performance still needs to be verified for datasets that combine UAV images with digital surface model (DSM). Adding DSM data to the U-net model could improve building extraction accuracy, but it also increases the learning difficulty of the network, because the U-net model parameters are learnt from optical RGB images. However, the use of fusion data for building extraction is worth attempting.

In response to the aforementioned issues, transfer learning was used to transfer two U-net models into a chain network, and a new chain network model, chain fully convolutional neural network (CFCN), was constructed to accurately extract the buildings. An improved U-net is the first component of the CFCN, which was achieved by adding a vortex module to select additional high-level building features as contextual information. This process will facilitate the extraction of buildings. The second component of the CFCN transfers a general U-net as a boundary constrained module to optimize building boundaries, hole filling, and "speckle" removal. The main objectives of this study were to (1) propose a new method, CFCN, for building extraction from high spatial resolution UAV images and DSM data; (2) improve multi-scale inference and enrich contextual information while using vortex modules; (3) refine and modify building boundaries through a chain convolutional neural network; and, (4) verify the performance of the CFCN utilizing a series of datasets from different building scenes.

This paper is arranged, as follows. In Section 2, an overview of the method, data preprocessing, building extraction (including building segmentation and boundary constraints), and post- processing are introduced. Section 3 provides analysis of the experimental method and experimental results. Section 4 is the discussion and Section 5 is a summary.

## 2. Materials and Methods

### 2.1. Overview of the Method

The input data for the proposed building extraction approach is a series of four-band fusion images that are composed of DSM and UAV images. Figure 1 illustrates the detection architecture of the proposed approach. It mainly includes the following three processes.

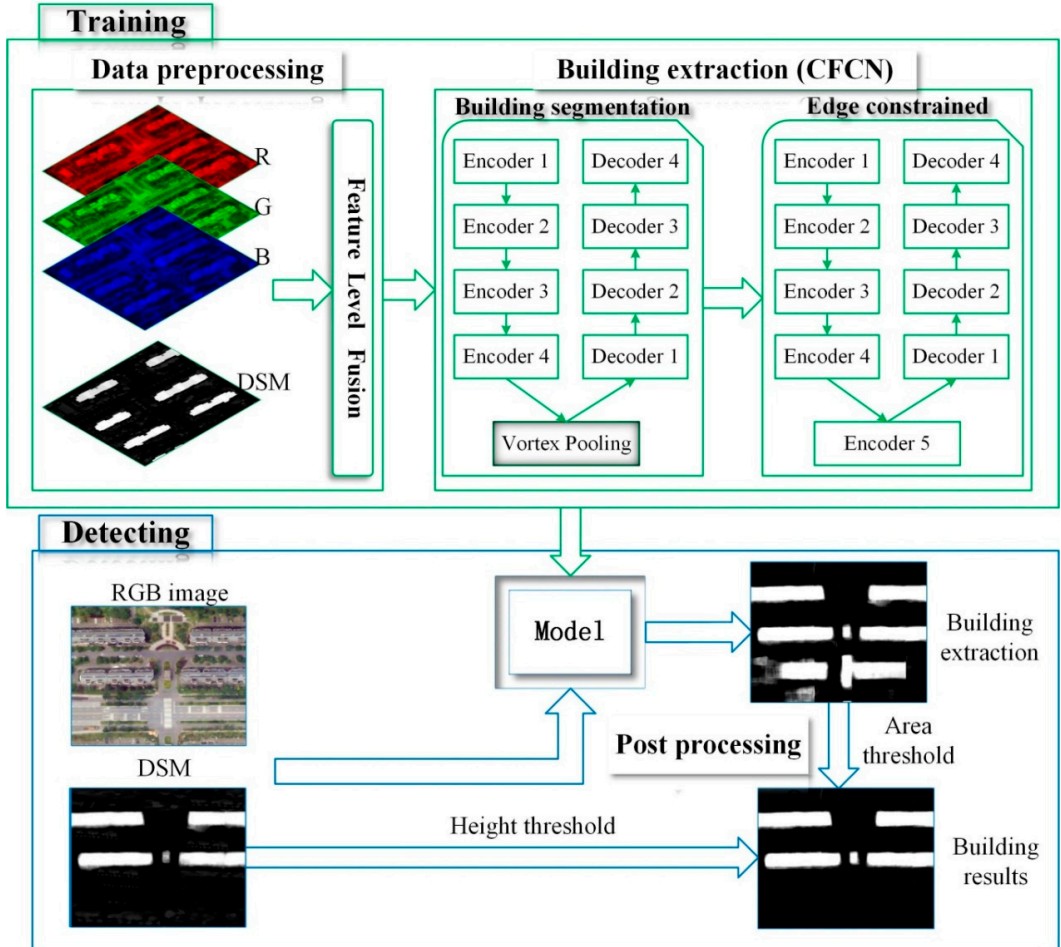

**Figure 1.** The process of the proposed building extraction.

Data preprocessing

First, the ground truth is prepared while using high spatial resolution UAV images and DSM. Subsequently, three-channel UAV images and one-channel DSM images are fused into a four-channel image using the Python version of the geospatial data abstraction library (GDAL2.4.2). The fused four-channel image and the ground truth image are sliced to generate a series of 256 × 256 patches and these are then fed into CFCN while using a GPU version of Tensor Flow 2.0 with a Keras application programming interface (API) for training and testing.

Building extraction

This section requires two chained convolutional neural networks for building segmentation and boundary constraints. First, the U-net network is optimized and a context module is added, that is, the U-net high-level feature encoding module is replaced with the vortex module to better utilize the context information. Thus, the building recall rate is improved and the segmentation results are optimized. Second, for the segmented buildings, where there are incomplete and unclear boundaries, a U-net network and the ground truth are used to perform boundary constriction, hole filling, and "speckle" removal on the segmented buildings.

Post processing

The area threshold and height threshold provided by DSM are applied to the building extraction results aiming to address the problem of misclassification of buildings, roads, and man-made landscapes.

## 2.2. Data Preprocessing

The UAV datasets that were used in this study were obtained on 5 May 2019 over the city of Yizheng, Jiangsu, China. Figure 2a shows part of the image, while the complete image is shown in experimental results, in order to better display local details of the image. The images were composed by three channels (RGB) with a ground resolution of 1.00 m, pixel resolution of 31,898 × 14,802, and an 8-bit radiometric resolution (Figure 3 UAV). The DSM was built while using Agisoft PhotoScan 1.4.5 with a resampling resolution of 1.00 m (Figure 2b). As the UAV image and the DSM have different features, data rescaling was applied while using normalization. In this process, the real valued numeric attributes are rescaled into the range of 0 and 1. A total of 7169 ground truth (GT) data were sampled from the UAV and DSM images. Each sample was labelled according to whether it is a building or not (Figure 2c). From the above dataset, 50% was allocated for training, 20% for validation, and the remaining 30% was used for a random test. A training dataset was used to train the DCNN, and this dataset contains labeled instance samples that were visible to the DCNN. The validation set was also clearly labeled for the DCNN. The validation set served two main purposes. First, it ensured that the model did not over fit and, secondly, it assisted in fine-tuning any necessary hyper-parameters. The test dataset for the DCNN model had labels that were not visible and, when the DCNN produced satisfactory results according to the training and verification sets, its performance was evaluated on the test set. In this study, the test set was used to ultimately verify the accuracy of our DCNN.

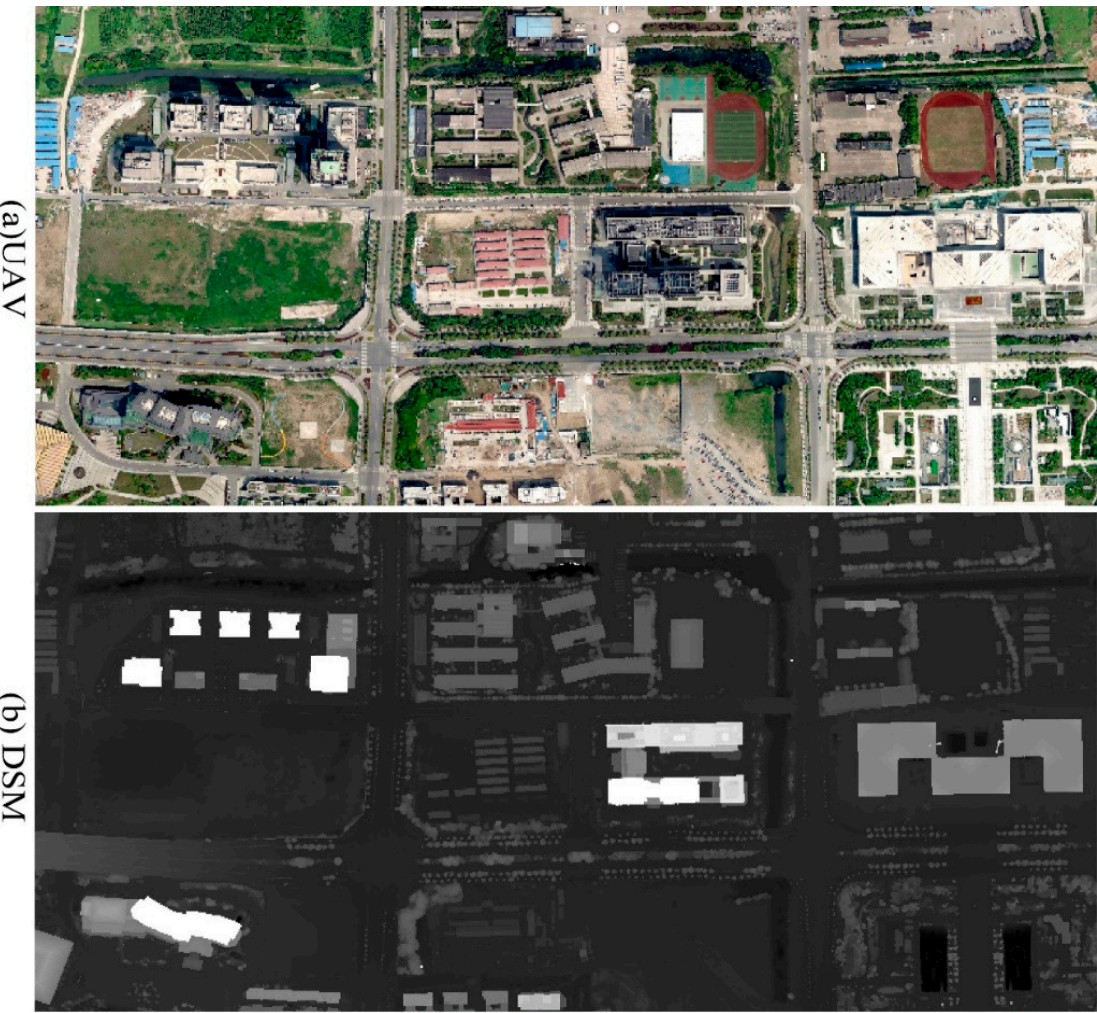

**Figure 2.** *Cont.*

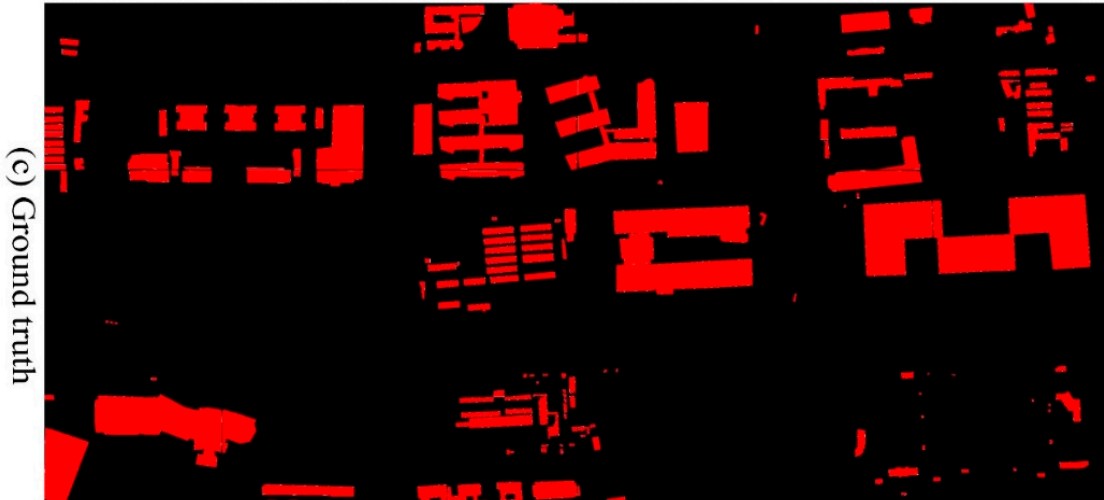

**Figure 2.** The datasets used in this study (**a**) unmanned aerial vehicle (UAV) image, (**b**) digital surface model (DSM), and (**c**) ground truth.

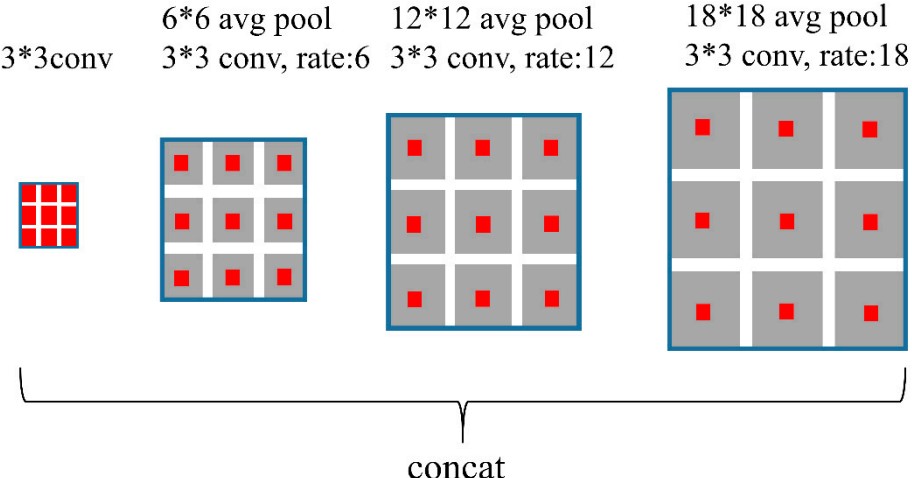

**Figure 3.** Vortex module in building segmentation.

## 2.3. Building Extraction

The CFCN consists of two full convolutional networks that form a chain network, one of which is a modified U-net with vortex modules to achieve coarse segmentation of buildings and the other is a U-net for the building boundary constraint, hole filling, and "speckle" removal.

### 2.3.1. Building Segmentation

The spatial information of the objects has an important influence on improving the segmentation accuracy in the process of building segmentation. The vortex module makes full use of context information by describing local interactions between neighborhood locations and the near-object region generally contains more semantic information, thus achieving better feature representation [37]. The algorithm of the CFCN selects the appropriate expansion rate and pooled convolution kernel for the data set, unlike the vortex module in [37]. As shown in Figure 3, the vortex module uses a $6 \times 6$, $12 \times 12$, and $18 \times 18$ average pooling for the feature maps in parallel, and the pooling layer enables the network to reduce the amount of computation and obtain global context information. The small pooling can better express the building details, and the large pooling can help obtain coarse building context information. Finally, four convolution layers of parallel expansion ratios (1, 6, 12, and 18) are used, and the convolved four feature maps are aggregated.

Ronneberger et al. in 2016 first proposed the U-net [27] and it has been widely used as a backbone network in image segmentation fields [38]. The modified U-net uses the vortex module to acquire multi-scale information and integrate global features for high-level abstract features in the encoding process (Figure 4). The vortex module output features are up-sampled during the decoding process, and the high-level semantic features that are obtained by the encoder are restored to the original image size. When compared to FCNs and SegNet, the improved U-Net makes full use of the context information of high-level semantic features and it uses skip connection, instead of directly supervising and loss backpropagation on high-level semantic features, which ensures that the final restored feature map not only incorporates more low-level features, but also allows for features of different scales to be fused, thus enabling multi-scale prediction and deep supervision [37–39].

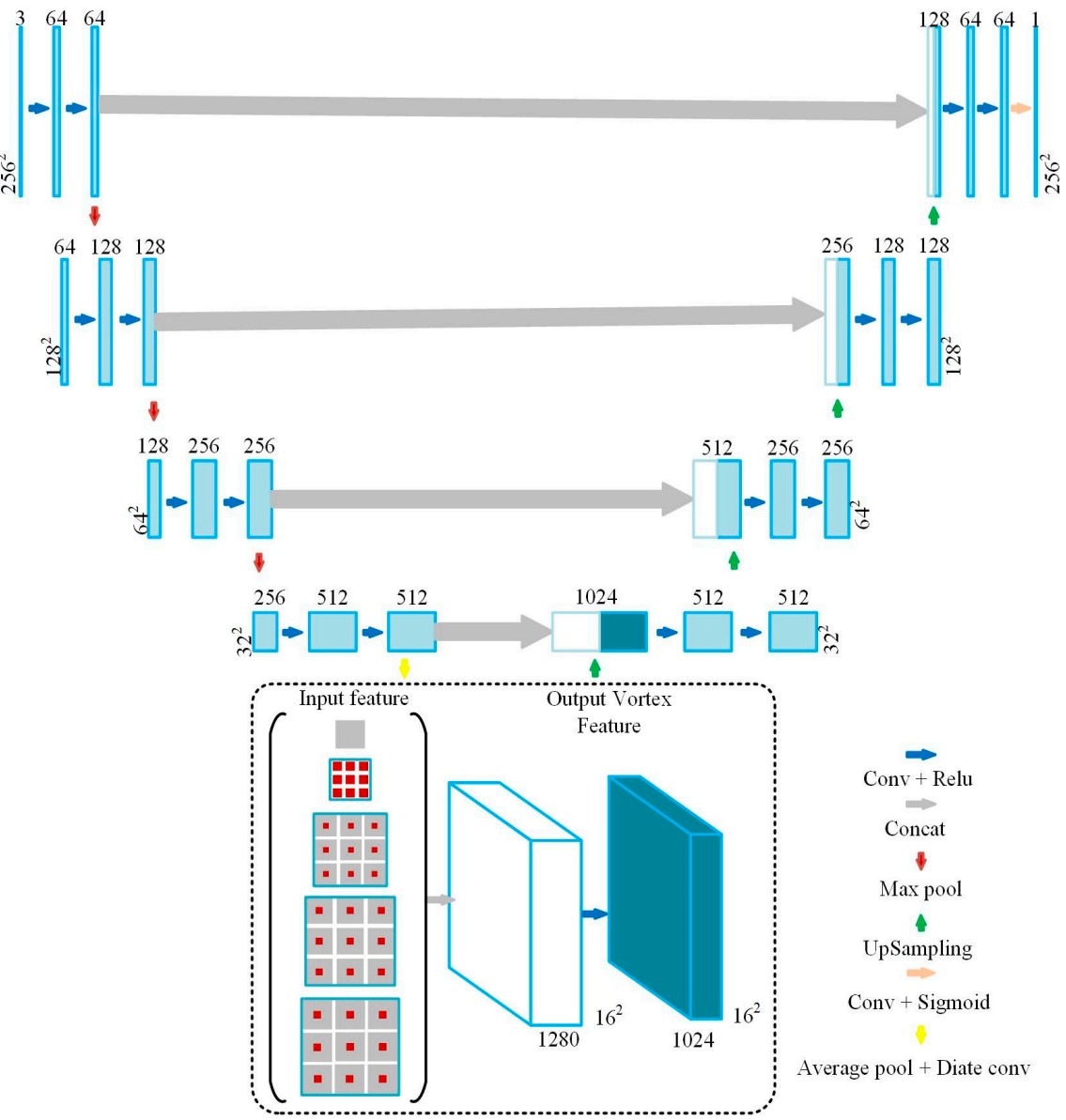

**Figure 4.** Architecture of the building segmentation in CFCN.

## 2.3.2. Boundary Constraint

The boundary constraint network is a general U-net network, which is an encoding-decoding process. Down-sampling four times in the encoder is a process of feature extraction. The low-level extract features are more inclined to form the basic unit of the image. The high-level extract features

approximately represent the semantic information of the image; the decoding process is the reverse operation of the encoding process. Up-sampling for four times restores the size of the original image, and Figure 5 shows its architecture. Input data of the boundary constraint network is the coarse building feature maps, which are the outputs mentioned in Section 2.3.1. A chained U-net network corrects the coarse building feature map. In this process, the U-net network will train the coarse building features to compensate for the incompleteness of the boundary in the extraction of coarse building features, the existence of holes in the building surface, and the "speckle" noise.

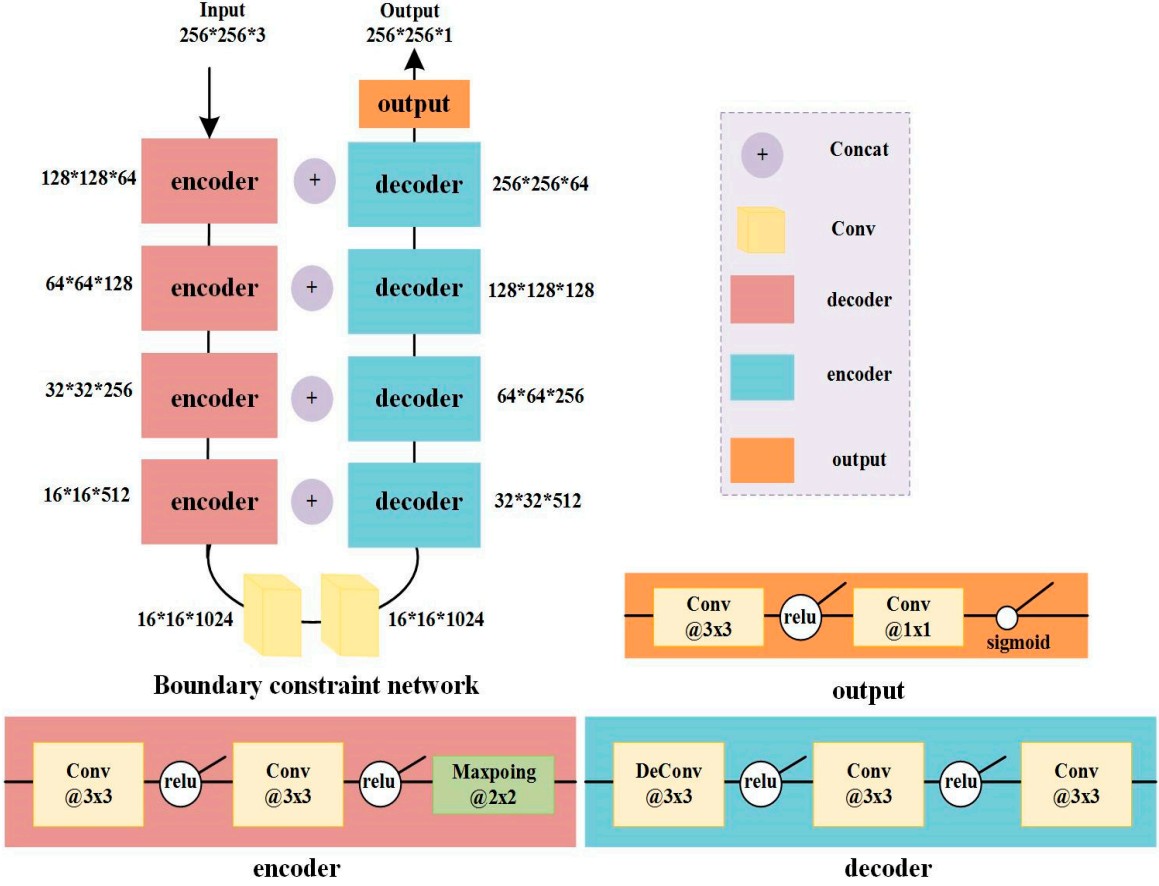

**Figure 5.** Architecture of the boundary constraint in CFCN.

## 2.4. Post Processing

After obtaining the original results of the building extraction, some small areas of island and some hardened ground (like roads), which are similar to buildings, will be preserved. Therefore, two post processing steps are adopted to filter the initial results of building extraction.

### 2.4.1. Height Filtration

A height filtration threshold $\tau$ is used to refine irrelevant low-lying objects, such as fence, roads, and hardened ground, $\tau$ should be large enough to eliminate possible interference, but small enough to retain low-rise buildings. After selecting $\tau$, if the absolute height obtained from the DSM is less than $\tau$, the corresponding pixels are deleted. When the height is 2.8 m, effectively eliminating irrelevant low-lying objects and deleting false buildings can achieve a better balance (Figure 6). In this study, the height threshold $\tau$ is set to 2.8 m higher than the surrounding mean ground height.

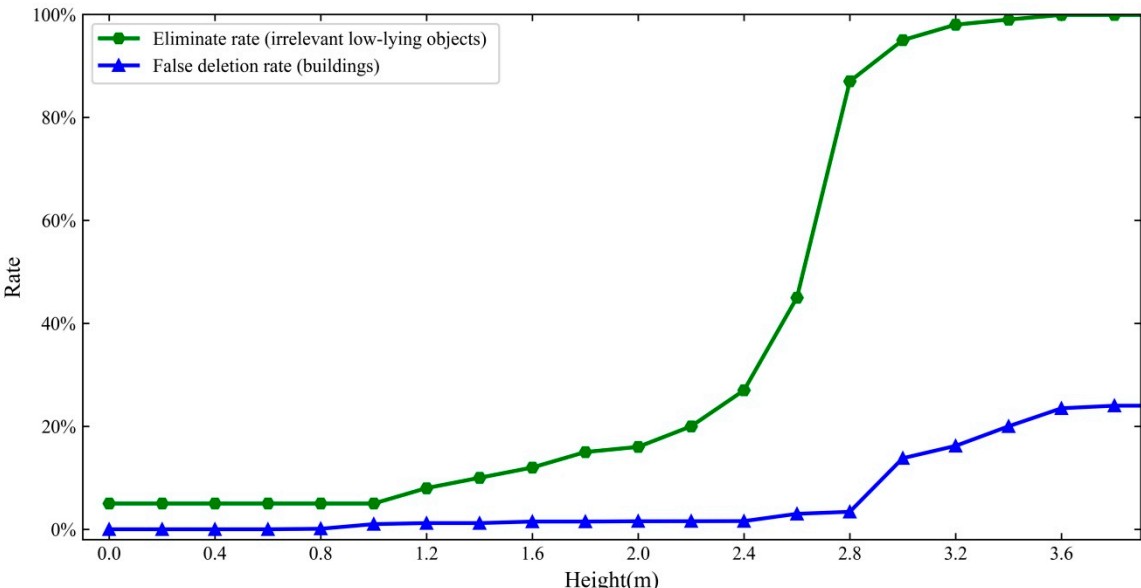

**Figure 6.** Error curve for height threshold in post-processing.

### 2.4.2. Area Filtration

In addition to some unrelated low-rise objects, some irrelevant small islands and small patches may be preserved in the extraction results. When considering that each building generally occupies a certain amount of area, area threshold $\varepsilon$ is used to eliminate these small islands. Connectivity analysis was carried out on the results of building extraction before filtering, and the connected regions were divided into separate patches. Subsequently, remove all of the patches smaller than $\varepsilon$ from the initial extraction result.

## 3. Experiments and Results

### 3.1. Experimental Design

#### 3.1.1. Experimental Setting

In this study, we compared the performance of U-net (input UAV image only), U-net add vortex module (Figure 4; input UAV image only, named U-net-V), U-net add vortex module (Figure 4; input UAV image and DSM, named U-net-VDSM), and CFCN (Figures 4 and 5; input UAV image and DSM). All of the input parameters for U-net, U-net-V, U-net-VDSM, and CFCN were selected through experience and many experiments with various values and choosing the ones with the best performance. Data augmentations were applied to the training set images to avoid network over fitting and improve model precision and robustness. Each input image was cropped to create a sequence of $256 \times 256$ pixel patches with an overlap of 128 pixels. The patches were randomly rotated, flipped, and translated while using the Keras API. The CFCN network were trained by the cross entropy loss function using stochastic gradient descent with a momentum optimizer and mini-batch size of 32 with the momentum of 0.9. The weight decay of 0.0005. The total iteration epochs were set 30,000 and the learning rate was set with 0.0005 and it was stepped down 10 times every 100 epochs. The CFCN network was implemented while using Tensor Flow 2.0 on an NVIDIA GeForce GTX Titan X GPU with 64 GB of memory under CUDA 9.0.

#### 3.1.2. Evaluation Metrics

Four most common evaluation metrics, including precision (correctness), recall (completeness), $F_1$ score, and mean intersect over union (IoU), were employed to evaluate the performance of U-net,

U-net-V, U-net-VDSM, and CFCN. Precision is used to measure how many true target-pixels in detected target-pixels and it is obtained as:

$$Precision = \frac{True\ positives}{True\ positives + False\ positives}$$

Here, recall or completeness, defined as:

$$Recall = \frac{True\ positives}{True\ positives + False\ negatives}$$

Represents the fraction of how many true target-pixels are identified in detected target-pixels. $F_1$ score is twice harmonic value of recall and precision and it is given by:

$$F_1 = 2 \times \frac{precision \times recall}{precision + recall}$$

IoU is used to measure the overlap rate of detected buildings and labeled buildings and it is defined as:

$$IoU = \frac{target \cap detected}{target \cup detected}$$

*3.2. Experimental Results*

In suburban areas, the buildings are relatively low, partially attached, and covered by trees (Figure 7a). U-net can identify most of the buildings. However, there are omissions in the tiny buildings, the "speckle" is serious, and some large buildings have holes. The extraction performance for the buildings on the edge is poor and the buildings show a serious adhesion phenomenon. In addition, tall trees partially cover the buildings, which lead to the inconsistency between the identified building boundaries and the actual building boundaries and it causes the absence of building corners. U-net-V could extract more buildings, the boundary was better than U-net, and "speckle" and holes reduced. U-net-VDSM could not only improve the building recall rate more effectively and get better building extraction at the image edge, but also effectively reduce the misclassification of hardened roads and suburban low-rise bungalows, and enhance the extraction accuracy of buildings. However, the "speckle" increased when U-net-VDSM was used to extract buildings; CFCN basically eliminated the "speckle" and building holes that occurred when the U-net-VDSM model was used to extract buildings. Constraining the boundary of buildings, which effectively reduced building adhesion, optimized the boundary information of buildings.

In an urban area, the building layout was regular and easy to extract (Figure 7b). U-net could extract the vast majority of buildings. However, there were holes in the extraction of large buildings, such as shopping malls and factories, because the buildings were relatively regular and the "speckle" was little. Building identification by U-net-V was comparable to that by U-net, but the boundaries were more accurate than in U-net, and the holes are reduced. The shapes of buildings that were extracted by U-net-VDSM were more complete, which effectively reduced the possibility that hardened roads were misclassified into buildings, but there was more building adhesion and more "speckle". CFCN could drastically reduce the number of holes, and correct the building boundary to a very precise degree, effectively reducing building adhesion.

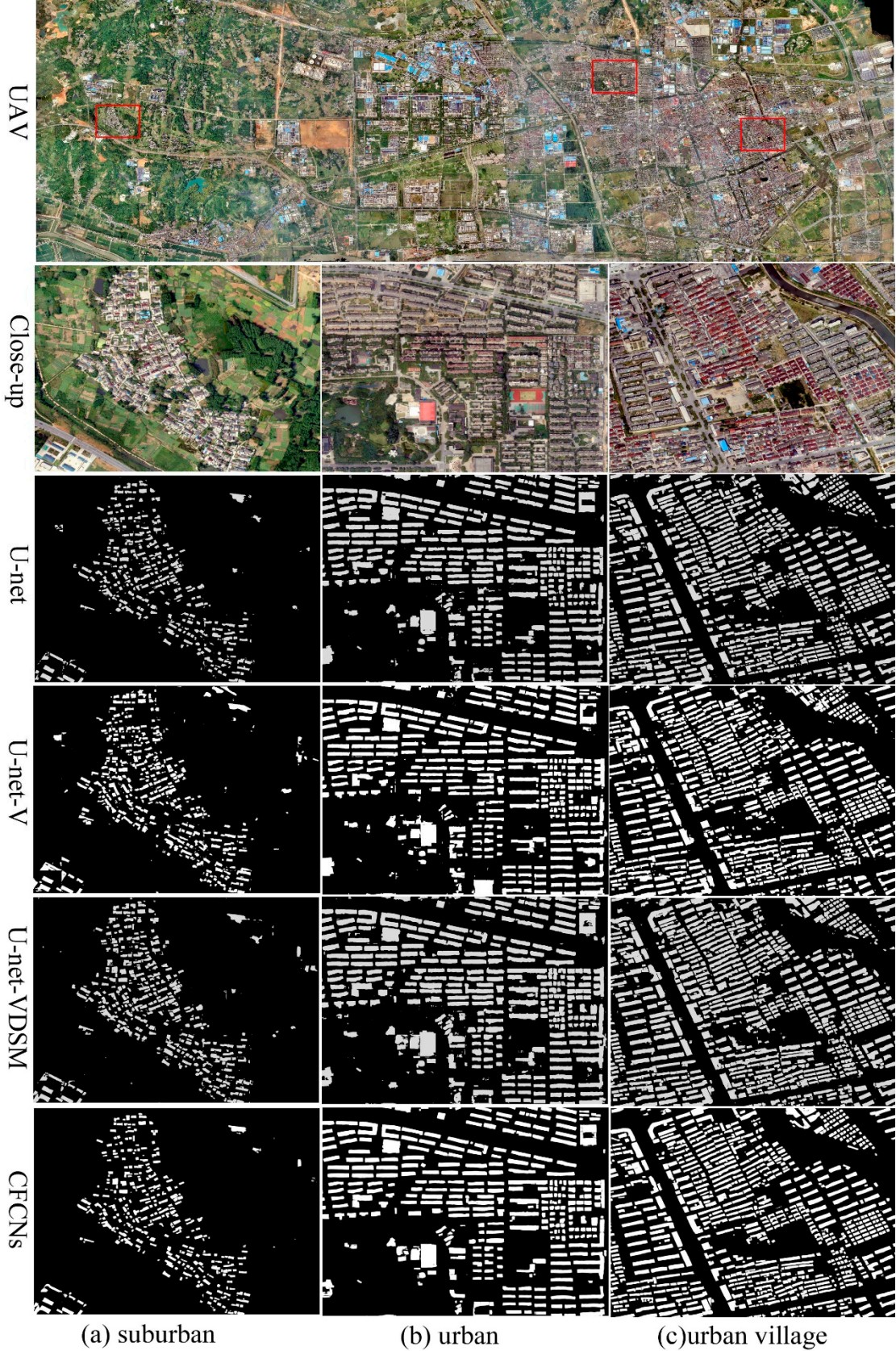

**Figure 7.** Close-up views of building extraction maps produced by different CNNs based on different scenes: (**a**) suburban; (**b**) urban; and, (**c**) urban-village. *Note: CFCN maps have not been post-processed*.

The buildings were low, dense, and contiguous, with adhesions and shades in an urban village area (Figure 7c). U-net could extract most buildings. However, a small number of buildings with large areas had holes, and there was "speckle" around low-rise and dense buildings. In addition, the extraction performance of buildings in the edge areas of images was poor. Some roads were misclassified as buildings due to the presence of zebra crossings. Buildings in shaded areas were not detected, which caused the lack of building corners. Omissions appeared in building extraction, and the identified building boundaries were coarse, due to the low and dense buildings. U-net-V shows obvious improvement in identifying building details when compared with U-net. Especially for U-shaped buildings, the extraction results of U-net-V were better than those of U-net. The number of holes in buildings decreased and the extraction of building boundaries was better than that of U-net. U-net-VDSM has better extraction performance on buildings at the image edge, and the shape of the extracted buildings is relatively complete. However, the holes increased when extracting the larger buildings. The extraction of building boundaries with CFCN was more in line with the actual shape of the building, which provided more detailed information. It could also improve the extraction performance for the buildings covered by tall trees and correct the corner and boundary of shaded buildings. The phenomenon of building adhesion also greatly improved, basically eliminating the "speckle" and holes.

Table 1 shows that U-net-V had a larger recall rate than U-net by adding vortex module, and the recall rate increased from 93.24 to 95.43%, which indicated that the vortex module can make better use of the context feature of low and dense buildings in suburbs. Furthermore, adding DSM features also improved the recall rate, which increased the U-net-VDSM recall rate to 98.86%. This is because DSM can effectively reduce the probability of misjudging suburban hardened courtyards as buildings, and effectively improve the accuracy and recall rate of building extraction. It can be seen from the intersection over union (IoU) that U-net-V is slightly better than U-net. As the use of vortex module can increase the recall rate of low buildings, it presents certain advantages in identifying building boundary. The IoU value of U-net-VDSM was higher than that of U-net-V. On the one hand, DSM features increased the recall rate of low buildings and effectively reduced building corner absence. On the other hand, the addition of DSM features improved the accuracy of building extraction. The use of CFCN had a certain inhibitory effect on the low-rise buildings that were covered by tall trees in the suburbs. Moreover, CFCN could effectively confine the boundary of buildings, fill holes, and remove "speckle", and its IoU reached up to 96.23%, which effectively corrected the boundary of low and dense buildings in suburbs.

**Table 1.** Precision, recall, $F_1$ and IoU obtained using different deep convolutional neural networks (DCNNs) in suburban.

| Methods | Precision | Recall | $F_1$ | IoU |
|---------|-----------|--------|-------|-----|
| U-net | 92.14 | 93.24 | 92.69 | 89.62 |
| U-net-V | 93.82 | 95.43 | 94.62 | 90.53 |
| U-net-VDSM | 95.14 | 98.86 | 96.96 | 92.61 |
| CFCN | 97.25 | 98.67 | 97.95 | 96.23 |

Table 2 shows that the recall rate of U-net was 98.15% in the urban area. The recall rate of U-net-V was slightly increased to 98.74% by adding a vortex module. On adding DSM information, the recall rate of U-net-VDSM went up to 99.57%. This is because urban buildings have standardized distribution, obvious features, and can be easily identified. High recall rates and accuracy could be achieved, even when the U-net model extracted the buildings. It can be seen from IoU that U-net-V was slightly better than U-net. This is because the vortex module can increase the recall rate of buildings, and the vortex module effectively reduces holes of buildings with larger areas. U-net-V performed slightly better than U-net-VDSM in IoU. Although U-net-VDSM slightly increased the recall rate, the boundaries of buildings that it extracted were not clear enough, and the "speckle" increased. The IoU could reach

up to 96.43, which effectively corrected the building boundary in urban areas while using CFCN for boundary constraint, holes filling, and "speckle" removal.

**Table 2.** Precision, Recall, $F_1$ and intersect over union (IoU) obtained using different DCNNs in urban.

| Model | Precision | Recall | $F_1$ | IoU |
|-------|-----------|--------|-------|-----|
| U-net | 96.47 | 98.15 | 97.30 | 91.12 |
| U-net-V | 96.82 | 98.74 | 97.78 | 92.84 |
| U-net-VDSM | 96.94 | 99.57 | 98.24 | 92.23 |
| CFCN | 97.22 | 99.52 | 98.36 | 96.43 |

Table 3 shows that the U-net-V showed an increase in the building recall rate over the U-net from 93.41 to 94.62% by adding the vortex module, which indicated that the vortex module effectively utilizes the context feature of low and dense buildings in urban villages. Furthermore, adding DSM features can also improve the recall rate, and the U-net-VDSM recall rate can reach up to 98.75. This is because DSM can effectively reduce the probability of dividing the hardened road into buildings, which effectively improves the accuracy and recall rate of building extraction. It can be seen from IoU that U-net-V was slightly better than U-net. This is because the vortex module was used to increase the recall rate of low-rise and dense buildings in the urban village, and the vortex module has certain advantages for building boundary recognition. The performance of U-net-VDSM is better than U-net-V in IoU. Corner absence was effectively reduced, as DSM increased the recall rate of buildings at the edge of the image. However, there was an increase in building adhesion. Utilizing CFCN could rectify the lack of corners for buildings covered by trees. Moreover, it can also restrain the building adhesion that DSM causes, and the IoU can reach up to 95.76.

**Table 3.** Precision, Recall, $F_1$, and IoU obtained using different DCNNs in urban village.

| Model | Precision | Recall | $F_1$ | IoU |
|-------|-----------|--------|-------|-----|
| U-net | 92.67 | 93.41 | 93.04 | 89.31 |
| U-net-V | 92.74 | 94.62 | 93.67 | 90.74 |
| U-net-VDSM | 94.62 | 98.75 | 96.64 | 92.32 |
| CFCN | 95.35 | 98.62 | 96.96 | 95.76 |

## 4. Discussion

There are three reasons for the excellent performance of CFCN. First, the encoder part of CFCN is based on the improved network architecture of U-net, which can effectively solve the degradation problem and simplify the process of achieving multiple levels of contextual information. Second, the building recall rate can be effectively improved by fusing DSM information, especially for buildings on the edge of the input image. It is likely that elevation information provided by DSM allows for the network to correctly distinguish between the buildings and backgrounds. Three, the initial extraction of the building can be effectively solve the problem of coarse boundary, holes, and "speckle", which is equivalent to fine secondary correction of the preliminary extracted buildings, through a U-net chain convolution. The U-net and its improved model have been widely used in building extraction and have achieved near perfect efficiency [28,31,38]. For example, Xu et al. [40] used a modified U-net network model for building extraction, and its recall, precision, and F1 reached 94.12%, 96.21%, and 95.15%, respectively. As significant literature discusses the application of the U-net and its improved version in building extraction, this will not be discussed here. Instead, the following discussion focuses on the effect of fusing DSM data and adding U-net chains for building extraction.

### 4.1. Effects of Fusing DSM Data for Building Extraction

Figure 8b shows the extraction result of U-net-V. The building at the edge of the image is unclear, and there are missing boundaries and corners (Figure 8b red rectangle). U-net-VDSM effectively solves

the problem of building extraction in the edge of UAV image upon adding DSM information (Figure 8c red rectangle). For some smaller buildings, after fusing the DSM feature, U-net-VDSM can effectively improve the recall rate of tiny buildings. From Tables 1–3, it can be seen that the dataset associated with the elevation information (the fused DMS and UAV images) performed better building extraction in all of the cases. It is indicated that the accuracy of building extraction can be effectively improved by fusing auxiliary data, which is consistent with the research results in [2,41–45]. In addition, there were cases where the U-net-V method mistakenly divided roads into buildings, especially in the zebra crossing area of the T-junction or the intersection, which makes the misclassification phenomenon more serious (Figure 8b green rectangle). Through the elevation information provided by the DSM, the probability of roads being mistakenly divided into buildings can be greatly reduced (Figure 8c green rectangle). However, despite including the DSM information, some roads were still misclassified as buildings. Fortunately, this problem could be effectively solved through post-processing.

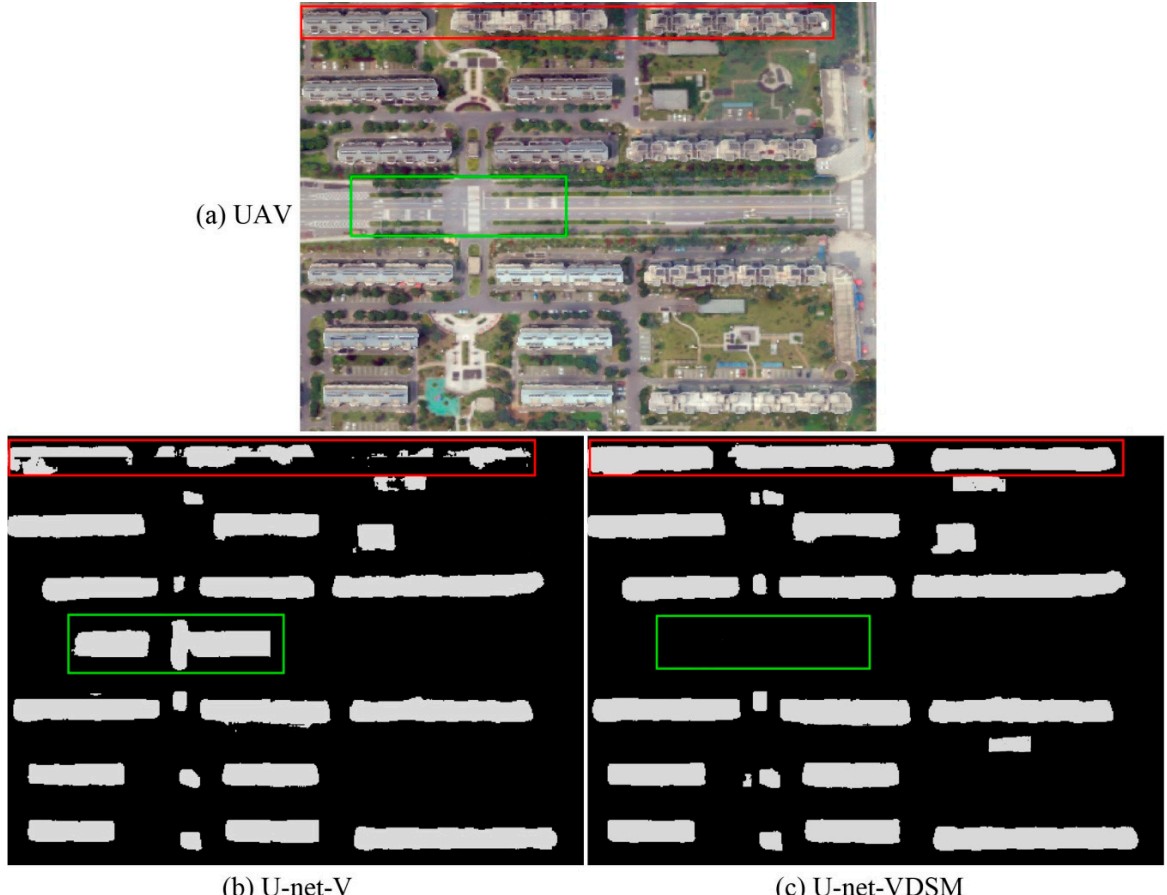

**Figure 8.** Building extraction results via different models. (**a**) UAV image; (**b**) U-net-V model; and, (**c**) input UAV image and DSM (U-net-VDSM) model.

### 4.2. Effects of Adding a U-NET Chain for Building Extraction

Extracting accurate building boundaries has always been a huge challenge in the field of remote sensing applications. In [46], a method for identifying and extracting buildings in high spatial resolution remote sensing images is proposed based on building shadow features and graph cutting algorithms, aiming to accurately extracting building contours. In [47], SVM is used to extract high spatial resolution remote sensing image building information, and, in [7], a morphological building index (MBI) is proposed for accurately extracting building contours. All of these methods are based on pixel features, which lead to problems in boundary blurring and incompleteness of the building, and extraction precision is low. Some scholars have applied CNN models to building extraction in order to improve

the accuracy of building extraction. In [48,49], the CNN model is applied to building extraction and the accuracy is greatly improved. However, problems of unclear and incomplete boundaries are prominent.

The CFCN is based on the U-net-VDSM, in which a U-net chain is added for boundary constraints, and it results in a more complete extracted building boundary. Figure 9b shows the results of building extraction by the U-net-VDSM method. It can be seen that the boundary of the building is coarse and there are many holes and "speckle". The building boundaries can be fixed to state-of-the-art performance by adding the U-net network chain to the boundary constraint (Figure 9c). It can be seen from Tables 1–3 that the IoU value of CFCN is significantly higher than U-net-VDSM after boundary constraints. This might benefit from the process of boundary correction of the chain convolutional network. As the building features are more pronounced after the preliminary building segmentation, the advantages of the deep convolutional constrained network can be utilized, and the visual effect of the constrained building boundary is remarkable (Figure 9c). However, the local features of some buildings are lost after the boundary of the building is constrained, because the convolutional network fills the small features of the building as holes (Figure 9b,c red circle).

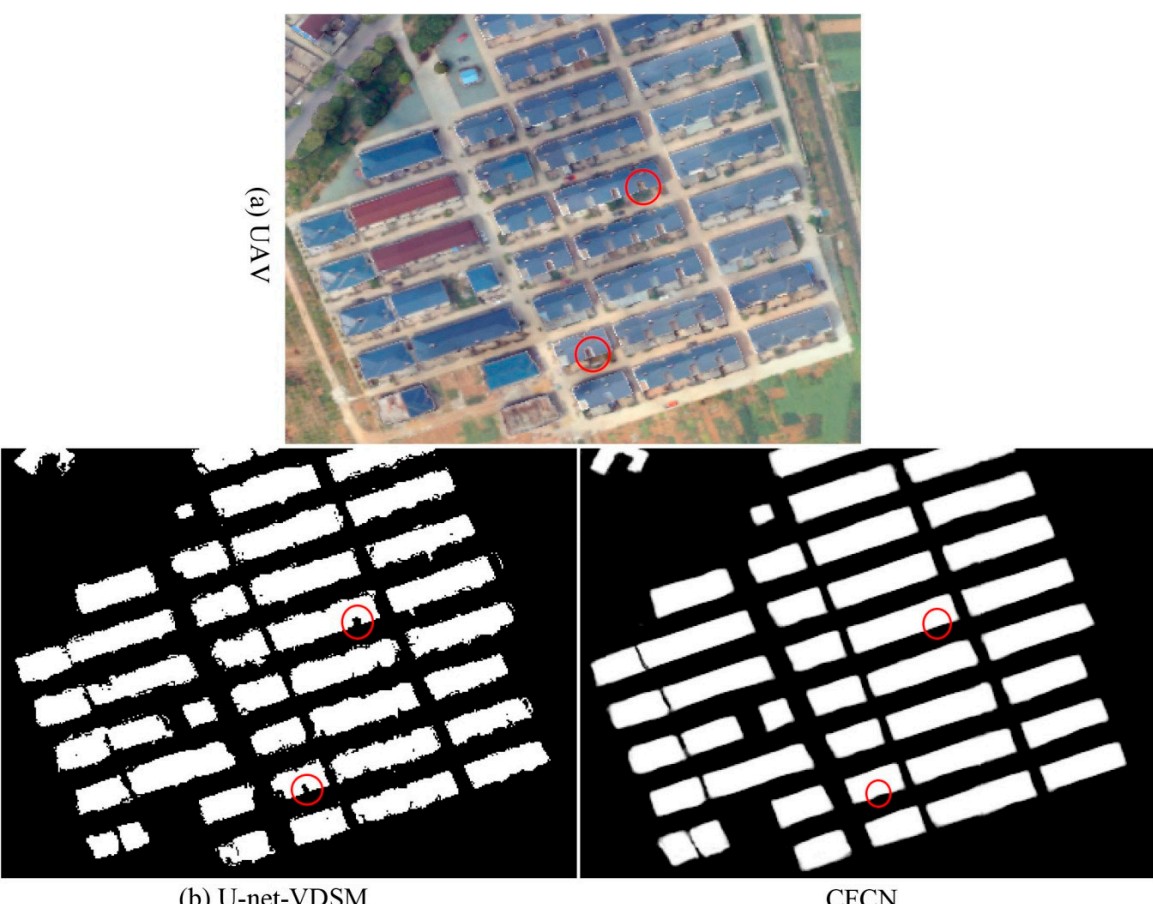

**Figure 9.** Comparison of results U-net-VDSM and CFCN. (**a**) UAV image; (**b**) U-net-VDSM; and, (**c**) CFCN.

## 5. Conclusions

In this paper, fused DSM and UAV images while using chain DCNNs that follow an encoder-decoder paradigm was proposed for the extraction of buildings, particularly for accurate building boundary correction. The building extraction process has three main components: (1) data preprocessing: fusion of DSM and UAV images through GDAL2.4.2 and semi-automatic generation of building ground truth samples; (2) building extraction: a U-net network to add vortex module is used for building extraction; the vortex module effectively increases the contextual information of high-level features of buildings

and improves the extraction of smaller buildings and a building boundary constraint network while using a general U-net network to achieve building boundary correction, filling holes and "speckle" removal; and, (3) post processing: the extracted buildings are filtered by area constraints and elevation information provided by DSM. The method is evaluated while using a high spatial resolution UAV image and DSM data that consists of image datasets with different surface elevations, shade cover, and building densities in suburban, urban, and urban village regions. Our proposed CFCN method achieved a competitive building recall of approximately 98.67%, 98.62%, and 99.52% in suburban, urban, and urban village areas, respectively. In particular, the results demonstrated that the method's IoU could reach approximately 96.23%, 96.43%, and 95.76% in suburban, urban, and urban village areas. This resulted in an estimated improvement of 6.61%, 5.31%, and 6.45% relative to the U-net method in suburban, urban, and urban village areas, respectively. The proposed method CFCN is effective, intuitive, and it will greatly contribute to building extraction while using state-of-the-art DCNN models. Future work will focus on establishing an end-to-end DCNN architecture and fusing more auxiliary data for extracting buildings in semi-supervised mode due to it the difficulties of obtaining ground truth data in most high spatial resolution remote sensing applications.

**Author Contributions:** Methodology, W.L. and M.Y.; Software, M.X. and D.W.; Visualization, M.Y. and Z.G.; Writing—original draft, W.L. and E.L.; Writing—review & editing, L.Z. and T.P.

**Funding:** This research was supported in part by the National Natural Science Foundation of China (No. 41590845, 41601405, 41525004 and 41421001), in part by a project funded Priority Academic Program Development of Jiangsu Higher Education Institutions(PAPD), in part by a grant from State Key Laboratory of Resources and Environmental Information System, in part by the Fund of Jiangsu Provincial Land and Resources Science and Technology Project (No. 2018054), in part by the Fund of Xuzhou Land and Resources Bureau Science and Technology Project (No. XZGTKJ2018001), and in part by the Fund of Xuzhou Science and Technology Key R & D Program (Social Development) Project (No. KC18139).

**Conflicts of Interest:** The authors declare no conflict of interest.

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
