# Peer review of "Accurate Building Extraction from Fused DSM and UAV Images Using a Chain Fully Convolutional Neural Network"

_remotesensing, doi:10.3390/rs11242912_

Round 1

Reviewer 1 Report

This is an interesting paper which focuses on building detection with use of DCNNs those apply four different approaches.

The main body of the proposed approach is based on the U-net structure.
I find the novelty is limited, since you apply already known method U-net for building detection from UAV image,
and use DSM for improving the results.
Although the DSM is used in fusing data, then we see it again at the post-processing step, with applying a hard threshold.
You also suggest to improve it with vortex pooling, but I see you only change the size of pools not more.

**you mention 'ground truth", is it training data (only for small samples) or ground truth (available for the whole dataset),
in case available for the whole dataset, then yours suggestion is based on the ground truth already,
so for use of your method in any other case may not work properly.
You say 50% was allocated for training, 20% for validation, and the remaining 30%, but if the boundary improvement is based on GT, how this will be possible.
Please give more details on this.

**Vortex pooling is introduced by [35], and the contribution/change by the author looks the change the size of pools as 6-12-18, instead of 3-9-27 as in [35].
The reason of selecting 2.8 meters is unclear,
it would be better to show another selected threshold with producing an error curve maybe.

** Line 271: You say Fig. 3a 3b and so on, but respective figure number is 6.

** A colored thematic result image is missing, this will show false +/- and true +/- .

** I find some related publications, but not listed in the paper, and easily could be found. The authors should much focus on related work for literature review.

** There are four approaches applied such as:
1. U-net (input UAV image only),
2. U-net add vortex module (input UAV image only, named U-net-V),
3. U-net add vortex module (input UAV image and DSM, named U-net-VDSM)
4. CFCNs (input UAV image and DSM).

here , it is not clear if only difference between 3 and 4 is size of pools. Because you mainly discuss with focusing U-net and extending the content with vortex, and suddenly we see CFCNs,
so please clearly indicate the difference between these two. And the training samples are all same for all approaches? You even don't mention CFCNs in the conclusions. You use the term DCNN as general which covers the all four approaches.
The contribution should be much clear in the text.

Author Response

    Thank you for your letter and for the reviewers’ comments concerning our submitted manuscript entitled “Accurate Building Extraction from Fused DSM and UAV Images Using a Chain Fully Convolutional Neural Network” (ID: remotesensing-630741). Your comments have proved very helpful with regard to our revisions and improvements of our paper, as well as providing insightful guidance for our future research. We have studied your comments carefully and have made the necessary corrections that we hope will meet with your approval. In the accompanying article, the revised portions of the text are marked in red. The principal corrections in the paper and our responses to the reviewers’ comments are detailed in the following.

The main body of the proposed approach is based on the U-net structure. I find the novelty is limited, since you apply already known method U-net for building detection from UAV image, and use DSM for improving the results. Although the DSM is used in fusing data, then we see it again at the post-processing step, with applying a hard threshold. You also suggest improving it with vortex pooling, but I see you only change the size of pools not more. Replay: Thank you for this insightful comment. We have rewritten goal and aims of this research, and replace "novelty" with "new" in line 127-137. You mention 'ground truth", is it training data (only for small samples) or ground truth (available for the whole dataset), in case available for the whole dataset, then yours suggestion is based on the ground truth already, so for use of your method in any other case may not work properly. You say 50% was allocated for training, 20% for validation, and the remaining 30%, but if the boundary improvement is based on GT, how this will be possible. Please give more details on this. Replay: Training dataset and Validation dataset contain labeled instance are known to the DCNN. The test set is a set unseen by the DCNN (i.e., labels are not visible). According to the reviewer’s suggestion. We have added some details in line 183-189. Vortex pooling is introduced by [35], and the contribution/change by the author looks the change the size of pools as 6-12-18, instead of 3-9-27 as in [35]. The reason of selecting 2.8 meters is unclear, it would be better to show another selected threshold with producing an error curve Replay: According to the shapes and size of the buildings in suburban, urban and urban village areas. Vortex pooling size chose 6-12-18 through many experiments with various values and choosing the ones with the best performance. Replay: According to the reviewer’s suggestion. We have added some description in line 241-242 and an error curve in page 9. Line 271: You say Fig. 3a 3b and so on, but respective figure number is 6. Replay: We are very sorry for our negligence. We have revised all the related errors in line 287, 300 and 314. A colored thematic result image is missing, this will show false +/- and true +/-. Replay: In order to show the overall effect of building extraction, we chose large scenes for display (as shown in Figure 7, each scenes has more than 300 buildings). In addition, the building is dense and contiguous. If use of colored thematic result image for display, there will be clustering phenomenon, the effect is not ideal. so we did not use the colored thematic result image. However, in order to show more details for the building extraction results, we using the small scenes close-up view which was discussed in section 4. I find some related publications, but not listed in the paper, and easily could be found. The authors should much focus on related work for literature review. Replay: According to the reviewer’s suggestion. We have added more related references in the manuscript. There are four approaches applied such as: U-net (input UAV image only), U-net add vortex module (input UAV image only, named U-net-V), U-net add vortex module (input UAV image and DSM, named U-net-VDSM) CFCNs (input UAV image and DSM).

here, it is not clear if only difference between 3 and 4 is size of pools. Because you mainly discuss with focusing U-net and extending the content with vortex, and suddenly we see CFCNs, so please clearly indicate the difference between these two. And the training samples are all same for all approaches? You even don't mention CFCNs in the conclusions. You use the term DCNN as general which covers the all four approaches. The contribution should be much clear in the text.

Replay: We are very sorry for our unclear description. CFCN is based on U-net-VDSM, adding a U-net chain for boundary constraints. The CFCN can make the extracted building boundaries more complete. According to the reviewer’s suggestion. We reorganized the Discussion and Conclusion, and added the corresponding description in line 127-133, 191 and 258-259. The training samples for 1 and 2 are the same; the training samples for 3 and 4 are the same. Through the comparison of 1 and 2, we can see the effect of building extraction after adding vortex module; the comparison of 2 and 3 can see the effect of building extraction after fusing DSM data; the comparison of 3 and 4 shows the effect of building extraction after adding the U-net chain for boundary extraction.

    We have tried our best to improve the manuscript and we have made many changes to the text that have not altered the essential content or framework of the paper, but have been intended to improve clarity, readability, and to reduce repetition and wordiness. These changes have not all been marked in red in the revised paper.
   We sincerely appreciate the effort and consideration of the Editors/Reviewers and we hope that the corrections will meet with your approval.
   Once again, we thank you very much for your insightful comments and suggestions.

Reviewer 2 Report

[Introduction 1] introduction is really abstract, it must include information about subject of research. The subject of research is evaluation of specific architecture proposed by authors. Therefore, introduction must include discussion about architectures applied for DSM and aerialphoto to detect building. Additionally, chain model is transfer learning, which is not discussed in introduction.

[Materials and methods 1] Figure 4 provides U-Net architecture extended with "vortex module". It would be correct to mention original source of U-Net using reference ("U-Net: Convolutional Networks for Biomedical Image Segmentation") and to mention, that authors extended it with vortex module.

Figure 5 is really similar to image of U-Net architecture presented in original image. Therefore, there can be copyright conflict. Strong recommendation of reviewer is to redraw or to present this architecture using some other way.

[Material and methods 2] Considering description of "Overview of method", the Python was applied. However, authors do not describe machine learning platform (Keras or clear TensorFlow, or some other platform?).

[Abstract] must be extended by experiment results (achieved precision).

[row 258] "F1 - score" is unclear - it can be read as "F1 minus score".

[Tables 1-3] "F1" neither "FI"

[Introduction 2] "novel" means something totally new, but article provides only modified U-Net. It is traditional approach among ML engineers to apply existing architecture and modules to construct solutions for their problems. According to content, the subject of research is to evaluate how vortex module and two-step architecture improve quality of segmentation. Recommendation is to rewrite goal and aims of research. 

[Comment 1] nowadays ML engineers traditionally apply normalization after convolution to decrease time for training. Was it applied? (Simply, clear U-Net does not include it, it was presented in SegNet. If - not, it is OK)

[Comment 2] reviewer does not found description about augmentation. It is important to mention, is it applied in research? Some researches show, that augmentation must be applied to improve precision and robustness of CNN.

Author Response

    Thank you for your letter and for the reviewers’ comments concerning our submitted manuscript entitled “Accurate Building Extraction from Fused DSM and UAV Images Using a Chain Fully Convolutional Neural Network” (ID: remotesensing-630741). Your comments have proved very helpful with regard to our revisions and improvements of our paper, as well as providing insightful guidance for our future research. We have studied your comments carefully and have made the necessary corrections that we hope will meet with your approval. In the accompanying article, the revised portions of the text are marked in red. The principal corrections in the paper and our responses to the reviewers’ comments are detailed in the following.

[Introduction 1] introduction is really abstract; it must include information about subject of research. The subject of research is evaluation of specific architecture proposed by authors. Therefore, introduction must include discussion about architectures applied for DSM and aerial photo to detect building. Additionally, chain model is transfer learning, which is not discussed in introduction. Replay: Based on the reviewer’s suggestion. We have added relevant discussion in introduction in line 127-133. [Materials and methods 1] Figure 4 provides U-Net architecture extended with "vortex module". It would be correct to mention original source of U-Net using reference ("U-Net: Convolutional Networks for Biomedical Image Segmentation") and to mention, that authors extended it with vortex module. Replay: According to the reviewer’s suggestion. We have added some references about U-Net and its applications in line 208-209. Figure 5 is really similar to image of U-Net architecture presented in original image. Therefore, there can be copyright conflict. Strong recommendation of reviewer is to redraw or to present this architecture using some other way. Replay: According to the reviewer’s suggestion. We have redrawn the figure 5. [Material and methods 2] Considering description of "Overview of method", the Python was applied. However, authors do not describe machine learning platform (Keras or clear TensorFlow, or some other platform?).

Replay: According to the reviewer’s suggestion. We have added the machine learning platform describe in line 152-153.

[Abstract] must be extended by experiment results (achieved precision). Replay: According to the reviewer’s suggestion. We have extended the abstract by experiment results. [row 258] "F1 - score" is unclear - it can be read as "F1 minus score". Replay: Thank you for this insightful comment. We have revised all "F1 - score" to "F1 score". Tables 1-3] "F1" neither "FI" Replay: We are very sorry for our negligence. We have revised all "FI" to "F1" in table 1-3. [Introduction 2] "novel" means something totally new, but article provides only modified U-Net. It is traditional approach among ML engineers to apply existing architecture and modules to construct solutions for their problems. According to content, the subject of research is to evaluate how vortex module and two-step architecture improve quality of segmentation. Recommendation is to rewrite goal and aims of research. Replay: According to the reviewer’s suggestion. We have rewritten goal and aims of this research. [Comment 1] nowadays ML engineers traditionally apply normalization after convolution to decrease time for training. Was it applied? (Simply, clear U-Net does not include it, it was presented in SegNet. If - not, it is OK) Replay: As the UAV image and the DSM have different features, the data rescaling was applied using normalization. But we do not use the normalization process after the convolution. We have added the description in line 178-180. [Comment 2] reviewer does not found description about augmentation. It is important to mention, is it applied in research? Some researches show, that augmentation must be applied to improve precision and robustness of CNN. Replay: Thank you for this insightful comment. We have added the augmentation part in line 261-265.

    We have tried our best to improve the manuscript and we have made many changes to the text that have not altered the essential content or framework of the paper, but have been intended to improve clarity, readability, and to reduce repetition and wordiness. These changes have not all been marked in red in the revised paper.
   We sincerely appreciate the effort and consideration of the Editors/Reviewers and we hope that the corrections will meet with your approval.
   Once again, we thank you very much for your insightful comments and suggestions.

Reviewer 3 Report

The research is very interesting, your improvement using DSM in the neural netkwok processing is well done.

I hope that the following tests add new skills

Author Response

    Thank you for your letter and for the reviewers’ comments concerning our submitted manuscript entitled “Accurate Building Extraction from Fused DSM and UAV Images Using a Chain Fully Convolutional Neural Network” (ID: remotesensing-630741). Your comments have proved very helpful with regard to our revisions and improvements of our paper, as well as providing insightful guidance for our future research. We have studied your comments carefully and have made the necessary corrections that we hope will meet with your approval. In the accompanying article, the revised portions of the text are marked in red. The principal corrections in the paper and our responses to the reviewers’ comments are detailed in the following.

    The research is very interesting; your improvement using DSM in the neural network processing is well done. I hope that the following tests add new skills

    Reply: Thank you for your positive assessment. As you suggested. Our future work will focus on establishing an end-to-end DCNN architecture and fusing more auxiliary data for extracting buildings in semi-supervised mode.

    We have tried our best to improve the manuscript and we have made many changes to the text that have not altered the essential content or framework of the paper, but have been intended to improve clarity, readability, and to reduce repetition and wordiness. These changes have not all been marked in red in the revised paper.
   We sincerely appreciate the effort and consideration of the Editors/Reviewers and we hope that the corrections will meet with your approval.
   Once again, we thank you very much for your insightful comments and suggestions.

Round 2

Reviewer 1 Report

The authors give high effort to improve the paper, and the paper has good and useful content. But I am sorry that I still think the novelty is limited since the contribution to Unet is limited.